

# Building interpretable models for polypharmacy prediction in older chronic patients based on drug prescription records

Simon Kocbek[1,2,3], Primoz Kocbek[4], Andraz Stozer[5], Tina Zupanic[6], Tudor Groza[1,7] and Gregor Stiglic[4,8]

[1] Kinghorn Centre for Clinical Genomics, Garvan Institute of Medical Research, Sydney, NSW, Australia
[2] Advanced Analytics Institute, Faculty of Engineering and IT, University of Technology, Sydney, New South Wales, Australia
[3] Department of Computing and Information Systems, University of Melbourne, Melbourne, Victoria, Australia
[4] Faculty of Health Sciences, University of Maribor, Maribor, Slovenia
[5] Institute of Physiology, Faculty of Medicine, University of Maribor, Maribor, Slovenia
[6] Healthcare Data Center, The National Institute of Public Health of the Republic of Slovenia, Ljubljana, Slovenia
[7] St Vincent's Clinical School, Faculty of Medicine, UNSW Sydney, Sydney, NSW, Australia
[8] Faculty of Electrical Engineering and Computer Science, University of Maribor, Maribor, Slovenia

Corresponding author
Simon Kocbek, skocbek@gmail.com

## ABSTRACT

**Background**. Multimorbidity presents an increasingly common problem in older population, and is tightly related to polypharmacy, i.e., concurrent use of multiple medications by one individual. Detecting polypharmacy from drug prescription records is not only related to multimorbidity, but can also point at incorrect use of medicines. In this work, we build models for predicting polypharmacy from drug prescription records for newly diagnosed chronic patients. We evaluate the models' performance with a strong focus on interpretability of the results.

**Methods**. A centrally collected nationwide dataset of prescription records was used to perform electronic phenotyping of patients for the following two chronic conditions: type 2 diabetes mellitus (T2D) and cardiovascular disease (CVD). In addition, a hospital discharge dataset was linked to the prescription records. A regularized regression model was built for 11 different experimental scenarios on two datasets, and complexity of the model was controlled with a maximum number of dimensions (MND) parameter. Performance and interpretability of the model were evaluated with AUC, AUPRC, calibration plots, and interpretation by a medical doctor.

**Results**. For the CVD model, AUC and AUPRC values of 0.900 (95% [0.898–0.901]) and 0.640 (0.635–0.645) were reached, respectively, while for the T2D model the values were 0.808 (0.803–0.812) and 0.732 (0.725–0.739). Reducing complexity of the model by 65% and 48% for CVD and T2D, resulted in 3% and 4% lower AUC, and 4% and 5% lower AUPRC values, respectively. Calibration plots for our models showed that we can achieve moderate calibration with reducing the models' complexity without significant loss of predictive performance.

**Discussion**. In this study, we found that it is possible to use drug prescription data to build a model for polypharmacy prediction in older population. In addition, the

study showed that it is possible to find a balance between good performance and interpretability of the model, and achieve acceptable calibration at the same time.

# INTRODUCTION

Multimorbidity is becoming increasingly common, especially in older population. Despite the improvements in chronic disease treatment, the prevalence of multimorbid patients is still on the rise, although it is difficult to exactly define the multimorbidity (*Willadsen et al., 2016*). However, it is known that prevalence of multimorbidity increases with age (*Calderón-Larrañaga et al., 2018*). Polypharmacy or concurrent use of multiple medications by one individual is becoming another major health concern and is tightly related to multimorbidity. Especially in the older population, the number of concurrent health conditions is directly related to a number of medications prescribed, eventually resulting in polypharmacy (*Hajjar, Cafiero & Hanlon, 2007*).

Detecting polypharmacy from drug prescription records is not only related to multimorbidity, but can also point at incorrect use of medicines. According to estimates by the World Health Organisation (WHO) more than half of all medicines are prescribed, dispensed or sold inappropriately, and that half of all patients fail to take them correctly (*WHO, 2012*). In the scope of the third global patient safety challenge, WHO addresses three areas of medication-related harm—i.e., high-risk situations, polypharmacy and transitions of care (*Sheikh et al., 2017*). With the rapid introduction of the electronic health records (EHR), particularly at the primary healthcare level, it will be possible to effectively monitor and identify groups of patients or individuals at high risk for drug-induced or related health problems (*Molokhia & Majeed, 2017*). Additionally, linking different EHR repositories together (*Kocbek et al., 2016*) and solving challenges in capturing the data in electronic form (*Stiglic et al., 2017*) will allow further improvements of data driven techniques.

A great majority of studies on polypharmacy have focused on its potential negative consequences, e.g., nonadherence, interactions, and adverse drug reactions. Some researchers have also considered the effectiveness of interventions aimed at reducing polypharmacy, however, the factors and conditions leading to polypharmacy have received comparatively little attention. These factors can be broadly classified into four groups: (i) factors related to the health care system (e.g., life expectancy and novel therapies), (ii) factors related to patients (e.g., age and clinical conditions), (iii) factors related to physicians (e.g., guidelines and prescribing habits), and (iv) the interaction between patient and physician. In our study, we focused on medical therapy, more specifically on medications taken in the last three months in older patients with newly diagnosed chronic cardiovascular disease (CVD) type 2 diabetes mellitus (T2D) patients. *Kanta et al. (2016)* demonstrate high prevalence of non-adherence problem as well as polypharmacy
in patients with CVD and T2D where fears of drug toxicity are mentioned as a barrier to taking medicines.

Machine learning is becoming indispensable for solving problems in many disciplines, including healthcare. At the moment, we are witnessing the introduction of various machine learning approaches in different fields of healthcare that can help the professionals in improvement of diagnosis or prognosis and even displacing a lot of work done by radiologists and anatomical pathologists (*Obermeyer & Emanuel, 2016*). However, despite the ever-increasing prediction performance of the novel predictive modelling techniques, most of them still lack interpretability to offer actionable support for healthcare experts (*Holzinger et al., 2017*; *Stiglic et al., 2012*). Therefore, this study aims to offer more insight into balancing the interpretability and predictive performance of the predictive models in healthcare. More specifically, we evaluate different levels of interpretability offered by regularized logistic regression modelling to predict polypharmacy based on prescription data in CVD and T2D patients.

# DATA AND METHODS

## Study design and data source

Two separate nationwide data sources were available for this study. The first dataset contained drug prescription records collected in Slovenia from 2008 to 2016, while the second dataset contained Slovenian hospital discharge records (primary healthcare level) from 2006 to 2016. Both datasets included patient identification information to allow linkage of data between years 2008 and 2016. All the data was collected centrally by National Institute of Public Health covering the whole population of Slovenia, which presents an important advantage compared to decentralised datasets where data linkage is not possible. The "Transparent reporting of a multivariable prediction model for individual prognosis or diagnosis" (TRIPOD) (*Collins et al., 2015*) was followed.

## Study setting

A total of 94,475,895 prescription entries for all patients who were prescribed at least one medication for T2D or CVD were obtained covering 755,966 unique patients (i.e., 402,286 males and 349,892 females, while 3,788 patients contained different genders at different time points and were later removed). The raw data contained 14 variables including anonymised patient id, patient gender, patient's geographical information, drug identifier, and the patient's doctor information. The hospital discharge data contained 1,740,610 entries covering 526,087 unique patients, who were prescribed at least one medication for T2D or CVD in the time period between 2006 and 2016. The discharge data was provided in two different formats, depending on time period when it was collected. Both formats included general information about the patient (e.g., age or anonymised identifier) and the admission (e.g., year and date of the hospitalisation or main diagnosis). International Classification of Diseases, revision 10 (ICD-10) was used to define specific diagnosis. ICD-encoding contains codes for diseases, signs and symptoms, abnormal findings, complaints, social circumstances, and external causes of injury or diseases. The main difference between the two hospital record formats was the number of ICD-10 codes. Data

collected between 2006 and 2012 contained only primary and secondary diagnosis codes, while data collected between 2013 and 2016 contained primary and up to 19 additional diagnosis codes. To allow unbiased use of hospitalization data, only data from 2013 to 2016 was used in this study.

All records in both datasets were anonymised by the Slovenian National Institute of Public Health using the following three steps. First, a random identifier was assigned to each original patient identifier number to allow data linkage across both datasets. Second, no age for the patients was provided, instead the patients were divided into age groups of 5 years. For example, the age group 0 would contain patients with ages from 0 to 4 years, the age group 1 would contain patients with ages ranging from 5 to 9 years, etc. Finally, only the year and the month of the prescription were given.

Anatomical Therapeutic Chemical Classification System (ATC) codes were used to code the prescribed medications. ATC codes consist of up to 7 characters and provide the following information:

- L1: indicates the anatomical main group (one letter).
- L2: the therapeutic main group (two digits).
- L3: the therapeutic/pharmacological subgroup (one character).
- L4: the chemical/therapeutic/pharmacological subgroup (one character).
- L5: the chemical substance (two digits).

## Preprocessing of the data

First, we removed prescriptions for patients with different genders at different time points (manual inspection revealed an error in data) and prescriptions with no ATC codes (e.g., data entry errors or prescriptions of medical appliances). Next, for each ATC code, we also included the ATC3 code, i.e., a shorter L3 version of the full ATC code (e.g., for B01AA we would add B01).

Next, since the prevalence of polypharmacy increases by age, especially in older population aged 65+ (*Calderón-Larrañaga et al., 2018*), only patients born before 1960 (to consider the age group window) were included in the study. Since the dataset did not contain date of birth for patients, we used age groups to estimate dates of birth. For each patient, we calculated the maximum possible year of birth ($yb$) from the patient's age group ($ag$) and year ($y$) of when the prescription was issued:

$$yb(ag, d) = y - ag * 5.$$

Then we averaged all maximum possible years for each patient, rounded to the closest integer, and got the final approximate year of birth. The mean of year of birth for the 755,966 unique patients in our data was 1946.

Finally, we merged hospitalisation discharge data with prescription records.

## Electronic phenotyping

Electronic phenotyping is often described as the process of identifying patients with a medical condition or characteristic (*Banda et al., 2017*). In our work we had to identify patients with: (a) polypharmacy, and (b) newly diagnosed chronic CVD or T2D condition.

**Table 1   Selected ATC codes for CVD and T2D.**

| Condition | ATC | Description |
|-----------|-----|-------------|
| T2D | A10 | Drugs used in diabetes. |
| | B01AA, B01AC | Cardiac agents (excl. ACE inhibators). |
| CVD | C01, C04A | Antihypertensives. |
| | C02, C07 | Peripheral vasodilators. |
| | C08, C09 | E.g., Beta blocking agents, Calcium channel blockers. |

We defined polypharmacy as a concurrent use of at least five medications. Concurrent use was defined as all medications that were prescribed in three consecutive months (e.g., January, February, March).

As patients with a chronic condition $c$ we selected all those patients that were prescribed at least one medication for $c$ every three months, for a period of twelve months. Table 1 shows ATC codes for T2D and CVD. The latter were selected based on the recommendation by *Huber et al. (2013)*, which developed an updated and improved measure of patients' chronic disease.

As mentioned in 'Study setting', only data from 2013 until 2016 was used in the phenotyping stage. Figure 1 illustrates the electronic phenotyping steps. We selected January 2016 as the prediction time point (PTP) and filtered patients based on the chronic disease and polypharmacy conditions. For the former, we selected all non-chronic CVD and T2D that became chronic at PTP, and for the latter, we removed all patients with polypharmacy before PTP.

## Predictor and variables

Final datasets contained 678 and 1,225 predictor variables for T2D and CVD respectively. The T2D dataset contained 44.9% of positive cases, while 21.8% of patients were positive in the CVD dataset. The latter indicates an unbalanced dataset which represents additional challenge for predictive techniques, and we had to consider this fact when evaluating our models. The predictor variables were arbitrary selected form a 3-months window before PTP and consisted of age, gender, hospitalisation, and ATC, ATC3 and ICD codes. Hospitalisation, ATC and ICD codes were indicator values {0,1}, Age was numeric, while Gender was a dichotomous variable. Table 2 presents statistics for predictive variables used in the prediction model for both datasets. Note that Age represents mean age in years (calculated from estimated year of birth), $n$ for Male, Female and Hosp represents number of instances, while $m$ for #ATC, #ATC3 and #ICD represents number of variables with at least one positive instance.

## Predictive modeling

Advanced statistical methods were applied to find patterns in the datasets for both chronic conditions, and a model to predict polypharmacy complications for patients was built. As one of our goals was to build interpretable models to increase usability, we restricted
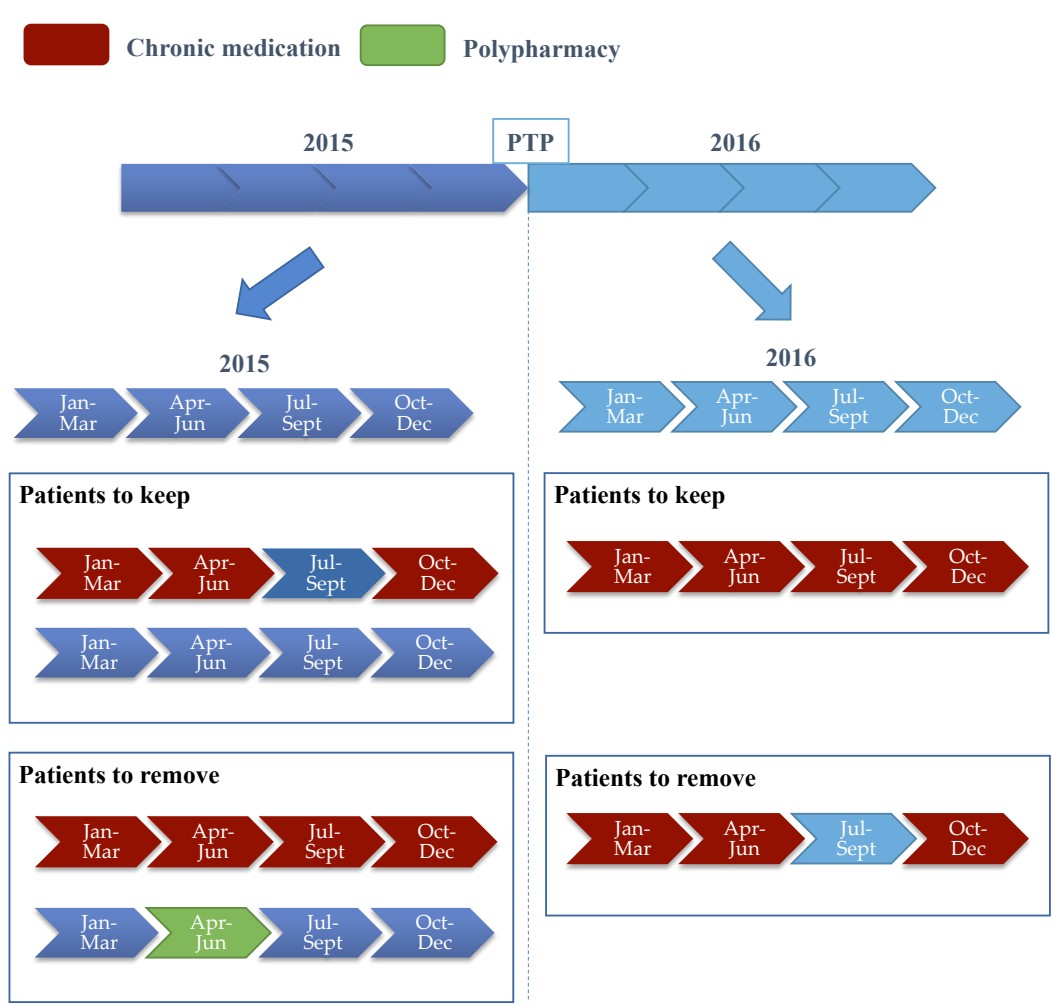

**Figure 1** **Summary of filtering chronic and polypharmacy patients where two years of prescription data are considered.** First, all data is partitioned into time periods of three consecutive months. Each three-month interval is used to: first, define the number of concurrent use of medications, and second, check for T2D or CVD medications. Next, the year before prediction time point (PTP) is used to remove all patients with previous polypharmacy (i.e., number of concurrent medications is higher than 4) or previous CVD or T2D chronic condition (i.e., at least one chronic medication is taken every 3 months). Finally, the year following PTP is used to select only the patients with a chronic condition, while polypharmacy in this year is used to define positive and negative patients.

model building to regularized linear models, where model complexity (dimensionality) can be tuned. The latter also helps in avoiding overfitting, a problem in machine learning where models do not generalise well. We experimented with both L1-norm (LASSO) and broader elastic net regularization, however, the latter resulted in more complex modes with no performance gain, therefore we report results only for LASSO. The generalized linear model via penalized maximum likelihood LASSO regularization was used as defined by

**Table 2 Summary table for predictor variables.**

| Variable | T2D | | CVD | |
| --- | --- | --- | --- | --- |
| | Pos ($n = 934$) | Neg ($n = 1,147$) | Pos ($n = 3,464$) | Neg ($n = 12,495$) |
| Age [95% CI years)] | 66.51 [66.03–67.00] | 65.34 [64.91–65.76] | 67.69 [67.42–67.99] | 65.72 [65.59–65.86] |
| Female [$n$ (%)] | 393 (42%) | 437 (38%) | 1,854 (54%) | 5,761 (46%) |
| Male [$n$ (%)] | 541 (58%) | 710 (62%) | 1,610 (46%) | 6,734 (54%) |
| Hosp [$n$ (%)] | 125 (13%) | 137 (12%) | 646 (19%) | 1,484 (12%) |
| #ATC [$m$ (%)] | 246 (36%) | 179 (26%) | 352 (29%) | 317 (26%) |
| #ATC3 [$m$ (%)] | 51 (8%) | 43 (6%) | 59 (5%) | 56 (5%) |
| #ICD [$m$ (%)] | 234 (35%) | 234 (35%) | 510 (42%) | 674 (55%) |

*Friedman, Hastie & Tibshirani (2010)*:

$$\min_{\beta_0, \beta} \frac{1}{N} \sum_{i=1}^{N} w_i l(y_i, \beta_0 + \beta^T x_i) + \lambda \|x_i\|_1,$$

where $i$ represents observations and it's negative log-likelihood contribution is noted as $l(y, n)$ with $w_i$ representing weights and tuning (shrinkage) parameter $\lambda$ controlling the overall strength of the penalty.

We further controlled the complexity of the model with the Maximal number of dimensions (MND) parameter with values from 10 to 100 in steps of 10, where the $\lambda$ parameter was optimized with respect to the internal 5-fold cross validation. More precisely, in the MND models the last $\lambda$ value before the number of predictor variables reaches MND is selected.

## Model validation

To evaluate our models, we focused on their predictive performance, which we describe in terms of *discrimination* and *calibration*. Discrimination measures the ability of a predictive model to separate outcomes, while calibration refers to the extent of the bias in the outcome of the model (*Harrell, Lee & Mark, 1996*). The discrimination can be measured by Area Under ROC Curve (AUC), i.e., the probability that the classifier will rank a randomly chosen positive case higher than a randomly chosen negative case. The ROC curve is plotted with Sensitivity or True Positive Rate (TPR) against the Fall-out or False Positive Rate (FPR), and AUC summarises the ROC curve into a single value by calculating the area of the convex shape below ROC. To obtain more details on predictive performance we also measured sensitivity, specificity, positive predictive value (PPV), negative predictive value (NPV), average number of selected features and percentage of positively predicted cases.

It is said that a model is "calibrated" when the predicted probability of a class matches the expected frequency of that class. Calibration can be visualized via a calibration plot, which plots class probabilities against those predicted by a single or multiple classifiers. In other words, calibration plots show observed proportion of events associated with a model's predicted risk, where the ideal calibration happens when both measures are equal. However, *Van Calster et al. (2016)* defined a calibration hierarchy, with the lowest level 1

(mean calibration or "calibration-in-the-large") and highest level 4 (strong calibration), where they presented a strong case for using moderate or level 3 calibration (i.e., the average predicted risk is equal to the actual average risk), which can be assessed via calibration plot. Van Calster et al. also proved that moderate calibration guarantees that clinically harmless decisions are made based on the model. The calibration plots presented in this study show both level 1, where the ideal case is a 45-degree line with a slope coefficient 1 and intercept 0 (*Steyerberg, Van Calster & Pencina, 2011*), and level 3 calibration.

To validate the predictive models, we performed repeated cross-validation. More specifically, 10-fold cross validation was used, whereby we randomly split data into 10 training/test sets for each model. We repeated this step 10 times, therefore we ended up with 100 experiments for each dataset. Instances in each fold were randomly selected. To evaluate and directly compare the models, the following two metrics were considered: AUC and Area Under the Precision Recall Curve (AUPRC). Similarly to AUC, AUPRC summarises the Positive Predictive Value (i.e, ratio of correctly classified positive values to the number of all instances classified as positive) over TPR curve into one number. AUPRC can often be more informative than AUC, especially for unbalanced datasets (*Saito & Rehmsmeier, 2015*), which was the case for the CVD dataset in this work.

Interpretability of models was measured by the number of selected variables in each experiment. We reported the following results: (a) number of all variables, and (b) number of all variables that were selected in all repetitions of the experiments for different MND values (i.e., stable variables). In addition, a medical doctor manually inspected all selected variables for MND = {10, 20, 50} to evaluate the interpretability (i.e., extracted knowledge in form of variables from the logistic regression models) of the models from the medical point of view.

## RESULTS

This section presents the results in terms of predictive performance, calibration and selected variables for both CVD and T2D datasets. All results are reported for both datasets with different MND values to observe the influence of model complexity on performance.

### Predictive performance

Box plots in Figs. 2 and 3 illustrate AUC and AUPRC values for both chronic diseases and different MND values ranging from 10 to 100 and an additional model with no dimension reduction (NDR).

One can observe stabilisation of both performance metrics in both datasets immediately after the MND is increased from 10 to 20. Even though a small increase in predictive performance can be observed when the complexity of the model increases, it is not significant, especially when MND increases to 50 and more selected variables.

More detailed predictive performance results including sensitivity, specificity, positive predictive value (PPV), negative predictive value (NPV), average number of selected features and percentage of positively predicted cases can be found in Supplemental Informations 1 and 2.

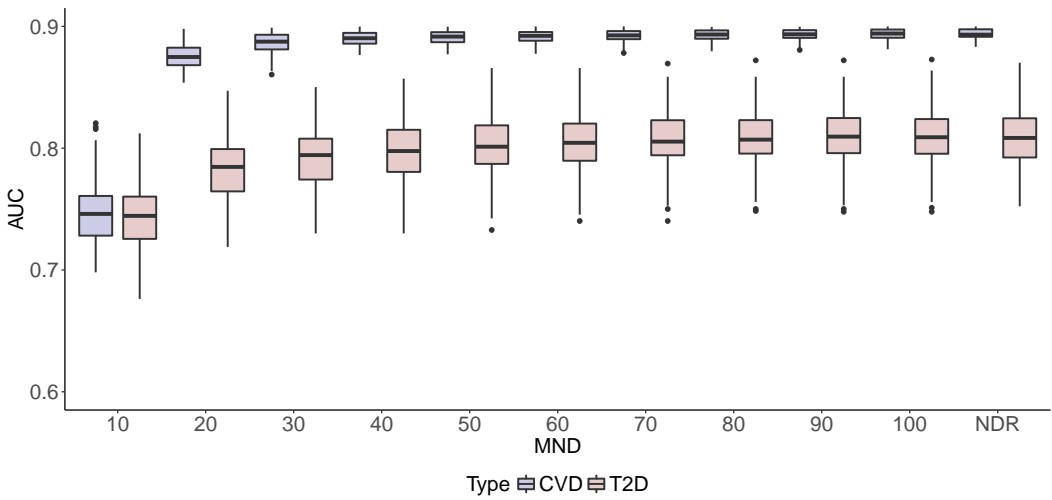

**Figure 2  Boxplots of CVD and T2D AUC values with 100 iterations at different MND values.**

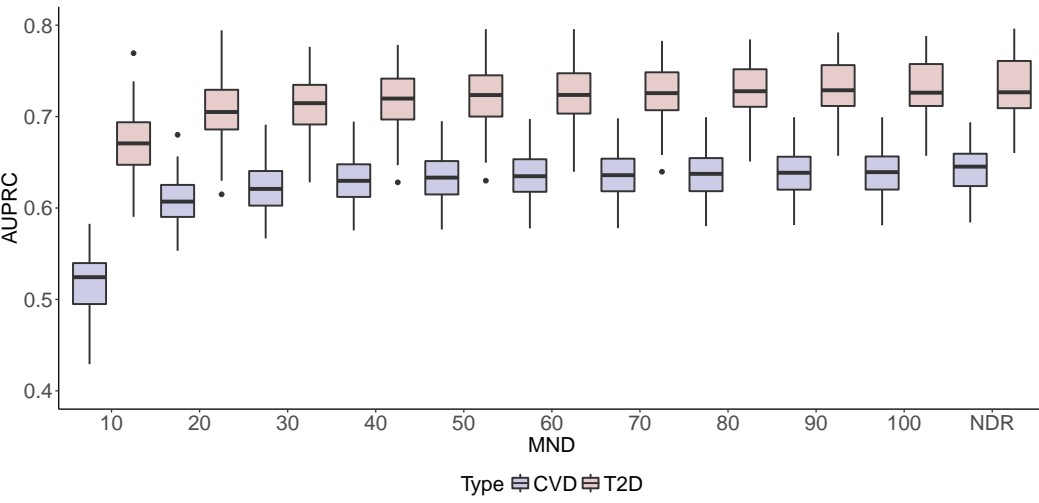

**Figure 3  Boxplots of CVD and T2D AUPRC values with 100 iterations at different MND values.**

## Calibration plots

Figure 4 presents the calibration plots for both chronic conditions. Due to space limitations, we show calibration plots only for MND = {10, 20, 50, 100, NDR}. The vertical axis of a calibration plot represents observed proportion of the class, while the horizontal axis represents the predicted probability.

Observing the calibration plots it can be noticed that the more complex models result in better calibration. However, the calibration improves significantly with the MND at 20 or higher.

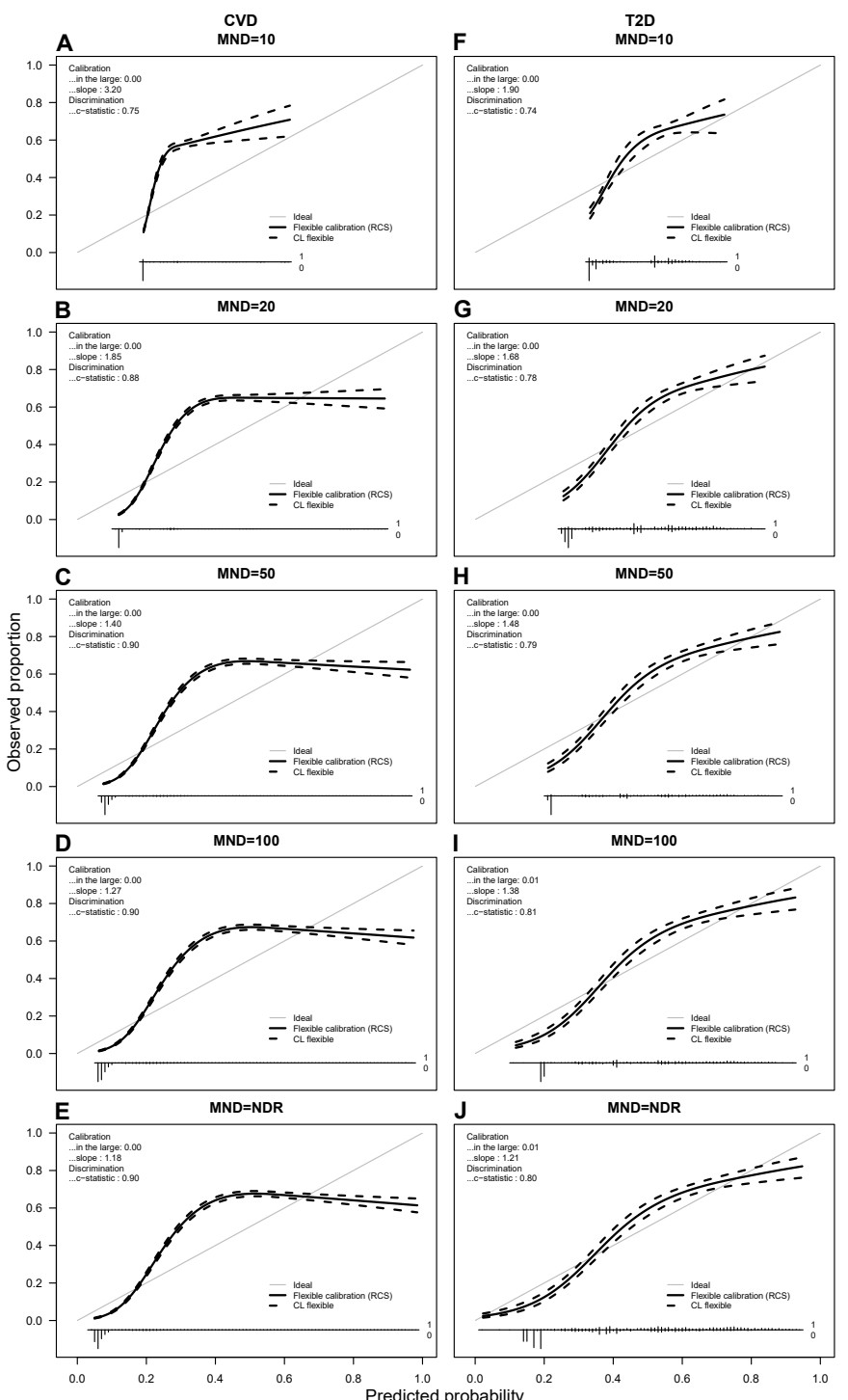

**Figure 4  Calibration plots for CVD (A–E) and T2D (F–J) for average probabilities and different MND values.** Predicted probabilities from each fold were saved and averaged over 10 repetitions. For each calibration plot in the upper left corner the intercept value (''in the large'') and slope is shown together with the AUC value or c-statistic. The main part of the plot is a flexible calibration curve based on restricted cubic splines, with a pointwise 95% confidence interval (dashed lines), followed by a case/non case histogram at the bottom.

**Table 3  Number of all and stable selected variables in all experimental repetitions.** Number of stable variables is presented in brackets.

| MND | 10 | 20 | 30 | 40 | 50 | 60 | 70 | 80 | 90 | 100 | NDR |
|---|---|---|---|---|---|---|---|---|---|---|---|
| CVD | 9 (4) | 21 (15) | 32 (17) | 47 (25) | 66 (29) | 75 (30) | 89 (31) | 107 (32) | 119 (32) | 125 (33) | 389 (43) |
| T2D | 11 (4) | 21 (12) | 37 (14) | 52 (14) | 71 (14) | 81 (14) | 91 (15) | 102 (15) | 126 (17) | 146 (18) | 352 (23) |

**Table 4  Ratio of all and stable selected variables in all experimental repetitions.** Ratio of stable variables is presented in brackets.

| MND | 10 | 20 | 30 | 40 | 50 | 60 | 70 | 80 | 90 | 100 | NDR |
|---|---|---|---|---|---|---|---|---|---|---|---|
| CVD | 0.03 (0.09) | 0.05 (0.35) | 0.08 (0.4) | 0.10 (0.58) | 0.13 (0.67) | 0.15 (0.7) | 0.18 (0.72) | 0.21 (0.74) | 0.23 (0.74) | 0.26 (0.77) | 1.00 (1.00) |
| T2D | 0.01 (0.17) | 0.01 (0.52) | 0.02 (0.61) | 0.03 (0.61) | 0.04 (0.61) | 0.05 (0.61) | 0.06 (0.65) | 0.07 (0.65) | 0.08 (0.74) | 0.08 (0.78) | 1.00 (1.00) |

## Selected variables

Due to space limitations, we list all selected variables for different experimental settings in Supplemental Informations 1 and 2, while Table 3 (Table 4) summarises the number (ratio) of (a) all variables, and (b) variables that were selected in all repetitions of the experiments for different MND values (i.e., stable variables) and NDR. Please note, that the number of all selected variables in an experiment can be higher than the experiment's MND parameter, since we repeat each experiment 100 times and its selected variables do not necessarily always overlap (which is the case for stable variables).

## DISCUSSION

In this study, we found that it is possible to use drug prescription data to build a model for polypharmacy prediction. Results on Figs. 2 and 3 (details in Supplemental Informations 1 and 2) show the maximum AUC (95% CI) values of 0.900 (0.898–0.901) and 0.808 (0.803–0.812) for CVD and T2D respectively, while AUPRC (95% CI) reaches maximum values of 0.640 (0.635–0.645) and 0.732 (0.725–0.739) for CVD and T2D respectively.

We see that the difference between AUC and AUPRC was lower for T2D compared to CVD. The CVD dataset is skewed towards negative class and consists of 21,8% positive cases (compared to 44.9% positive cases in T2D). It was shown, that AUC can be misleading in terms of the reliability of classification performance in imbalanced datasets, whereas AUPRC can provide an accurate prediction of classification performance, since they evaluate true positive amongst positive predictions (*Saito & Rehmsmeier, 2015*).

In addition, the study showed that it is feasible to find a balance between good performance and interpretability of the model. Figure 2 shows a slow decrease of the AUC performance for both datasets when complexity is controlled with MND. The difference with the maximum AUC values with dimensionality reduction is most notable with MND = 10, where AUC drops below 0.750 for both medical conditions. However, we can notice that increasing MND to 20, already improves performance significantly. Specifically, AUC of 0.875 (0.873–0.877) and 0.782 (0.777–0.787) is achieved for CVD and T2D respectively. The difference in AUC performance for MND = 20 compared to NDR is only 3% and 4% for CVD and T2D respectively, while Table 4 shows that only 5% and

1% of all variables have been kept for CVD and T2D respectively (35% and 52% of stable variables). The ratio between reduced complexity and decreased performance gets even smaller with higher MND values. For example, the maximum AUC value of 0.90 for CVD is achieved with MND = 50, with only 13% variables kept (67% stable variables).

Similarly to AUC in Fig. 2, results in Fig. 3 show how AUPRC changes when we control MND. The lowest performance was obtained for MND = 10, while already with MND = 20, AUPRC values reach 96% (0.607) and 95% (0.703) of the maximum AUPRC values for CVD and T2D, respectively.

The AUC and AUPRC results show that while medical experts are able to work with much less complex models when reducing MND, this does not mean that they have to significantly sacrifice performance of the model.

The study also showed that it is possible to achieve acceptable calibration when reducing complexity of the model. Calibration plots presented in Fig. 4 show the correlation between calibration and complexity of the model. *Van Calster et al. (2016)* recommend that strong calibration should be desirable in cases of individualized decision support, but often stimulates too complex models and might even be counterproductive in other cases. Our results confirm this recommendation as the models with the best calibration result in the highest number of selected variables, but without significant improvement of predictive performance. Further recommendation by *Van Calster et al. (2016)* introduces the so called moderate calibration defined by equality of the average predicted and the actual average risk. Moderate calibration can be observed in all models presented in our study.

Interpretation of selected variables by a medical doctor revealed that both CVD and T2D seem to be associated with polypharmacy independently of other medication (*Bjerrum et al., 1998*; *Jyrkkä et al., 2009*). Moreover, in our models for both CVD and T2D, a large proportion of selected medications suggest other clinical conditions that were reported in a review of nine studies to be associated with polypharmacy, e.g., depression, asthma, and gout (*Hajjar, Cafiero & Hanlon, 2007*). Further, some of the groups of drugs strongly suggest patient conditions other than well-defined diseases, such as declining nutrition and cognitive capacity that were also reported to be independently associated with polypharmacy (*Jyrkkä et al., 2011*). Concerning particular groups of drugs as predictors of polypharmacy, studies show large variation and this is further complicated by the fact that study settings differ as well (*Hovstadius & Petersson, 2012*). In a large study of an entire national population, *Hovstadius et al. (2010)* have found that the five most often prescribed drug groups in patients receiving polypharmacy were (listed in decreasing order occurrence) antibacterials, analgesics, psycholeptics, antithrombotic agents, and beta blocking agents. In our study, analgesics and psycholeptics were included as features across all models for both CVD and T2D. Beta-blockers were excluded in the CVD group, but included in all of the models for T2D. Interestingly, in our case antibacterials were included only in the CVD model and the antithrombotic agents were robust predictive features for the T2D model, whereas they were included in the CVD model only above MND = 50.

Among the features most consistently selected in our models for both CVD and T2D were psycholeptics, psychoanaleptics, and antiepileptics. It should be noted that particularly the latter are sometimes used for indications other than epilepsy, e.g., mood

stabilization. However, our finding corresponds with previous reports that people with mental health conditions and behavioral problems are at an increased risk for polypharmacy in general (*O'Dwyer et al., 2016*; *Peklar et al., 2017*) and that long-term use of some of the drugs from these groups carries the risk of metabolic dysregulation (*Gareri et al., 2006*), falls (*Peklar et al., 2017*) or even cognitive decline (*Jenkins, 2000*), which may precipitate a vicious cycle of receiving an increasing number of drugs.

Interestingly, compared with other studies reporting that women are more likely to receive polypharmacy (*Bjerrum et al., 1998*; *Haider et al., 2008*; *Qato et al., 2008*) and that increasing age is a key determinant of polypharmacy exposure (*Hovstadius & Petersson, 2012*; *Jyrkkä et al., 2009*; *Hajjar, Cafiero & Hanlon, 2007*; *Stewart & Cooper, 1994*), in our sample, no robust association was found between polypharmacy and gender, and age was a feature selected only for the CVD model above MND = 10. It is possible that some of the features selected in models yielding better prediction might not reflect worsening physical health and thus a greater biological need for polypharmacy due to true multimorbidity, but may reflect patient transfer to an institution or change in residency, since it has been found that some of the medications that were robustly selected in our models are more frequently reported for patients in residential, as compared to community group homes or those living independently (*O'Dwyer et al., 2016*) and that nursing home residents are at an increased risk for polypharmacy (*Vetrano et al., 2013*).

The addition of diagnosis data (inpatient ICD codes) showed little improvement in our models, both in terms of gain in AUC or AUPRC and in terms of selected features in the models. Our explanation for this outcome is twofold; firstly there were only 13.3% hospitalization associated with CVD cases and even less 12.6% with T2D cases, which gives us a sparse matrix with little information gain. Secondly, drug prescriptions are usually to some degree associated with diagnoses of hospitalizations especially for elder chronic patients, which lowers the information gain for these features in our models even more. Secondly, drug prescriptions are usually to some degree associated with diagnoses of hospitalizations (*Klarin, Wimo & Fastbom, 2005*) especially for older chronic patients, which lowers the information gain for these features in our models even more.

The present study has some limitations that should be taken into the consideration. First, the data was restricted only to patients with at least one prescribed medication for either CVD or T2D. Second, since the records contain only month and year of the prescription or hospital admission, this influences our definitions for concurrent use of drugs, polypharmacy and chronic disease. Third, due to age groups, we were able to only estimate years of birth. Finally, a set of ATC codes for CVD was based on previous work of *Huber et al. (2013)*.

## CONCLUSION

In this study we developed models to predict polypharmacy based on drug prescription and hospital discharge datasets. We focused on two common chronic conditions, i.e., CVD and T2D, since both are known to increase the risk of polypharmacy. Based on a centrally collected national prescription dataset, we defined and performed electronic phenotyping

of chronic CVD and T2D patients with/and without polypharmacy. We also measured how increasing interpretability of predictive models by decreasing the number of variables included in the final model influences their performance. The interpretability of predictive models is important for the application of the proposed model in practice, especially in the context of learning healthcare systems where models are continuously adapted.

In the future, we plan to investigate performance of our models on other diseases and apply deep learning (DL) algorithms (*Miotto et al., 2017*). With DL, we expect less interpretable models with increased performance. We believe that our work has potential to positively influence drug prescription practices as discussed in *Molokhia & Majeed (2017)*.

### Funding
This study was supported by the Slovenian Research Agency (research core funding No. P2-0057 and No. P3-0396), and AdFutura grant (11013-42/2017). The funders had no role in study design, data collection and analysis, decision to publish, or preparation of the manuscript.

### Grant Disclosures
The following grant information was disclosed by the authors:
Slovenian Research Agency: P2-0057, P3-0396.
AdFutura: 11013-42/2017.

### Competing Interests
The authors declare there are no competing interests.

### Author Contributions
- Simon Kocbek conceived and designed the experiments, performed the experiments, analyzed the data, prepared figures and/or tables, authored or reviewed drafts of the paper, approved the final draft.
- Primoz Kocbek performed the experiments, analyzed the data, prepared figures and/or tables, authored or reviewed drafts of the paper, approved the final draft.
- Andraz Stozer analyzed the data, authored or reviewed drafts of the paper, approved the final draft, medical interpretation.
- Tina Zupanic contributed reagents/materials/analysis tools, authored or reviewed drafts of the paper, approved the final draft.
- Tudor Groza approved the final draft.
- Gregor Stiglic conceived and designed the experiments, analyzed the data, authored or reviewed drafts of the paper, approved the final draft.

### Data Availability
The code is in the Source code.zip file. From Readme.txt:
Overview

- input data: preprocessed data in rds format (in directory data) - must be supplied
- output variable: poly
- numeric variable: YearBirthPerson, also GenderPerson treated as numeric
Glmnet output (directory output):
-output for results table: Name, Model, Fold, Seed, pmax, AUC, AUPRC, Threshold, Brier, Accuracy, Sensitivity, Specificity, PPV, NPV, NumSelected, Zero_pred (nr. of removed columns with cases), YLength, YtLength, YSum, YtSum, PercPos, Pred_Vars (predictor variables in model with sign)
-output for calibration plot: for first seed (Probability predictions, True value 0/1) for each MND
Aggregation table results (output directory reports)
- agregated selected metrics for CVD and T2D in separete files
- selected predictor variables with sign wrt MND for CVD and T2D in separete files
Figures (output directory figures)
-boxplots AUC for CVD and T2D wrt MND
-boxplots AUPRC for CVD and T2D wrt MND
-Calibrations plots for CVD and T2D wrt selected MND
Code order to run:
-glmnet_calib_res_atc_icd_CVD.r
-Results_table_atc_icd_CVD.R
-glmnet_calib_res_atc_icd_T2D.r
-Results_table_atc_icd_T2D.R
-calibration_plots_ATC_ICD.r

Due to the sensitive nature of data used in this study, supporting micro-data data cannot be made openly available. In case of large databases of health microdata, which are in Slovenia by Personal Data Protection Act of the Republic of Slovenia treated as sensitive data, microdata can be processed only at National Institute of Public Health's (NIPH) secure room, located at the premises of NIPH in Ljubljana. No export of such data to any other location by any means of transport is allowed. Therefore, data are not publicly available or cannot be transmitted to the third party other than researcher who signed the contract on non-disclosure of microdata. Individual requests for the data can be sent to the NIPH (Metka Zaletel; metka.zaletel@nijz.si).

## Supplemental Information

Supplemental information for this article can be found online at http://dx.doi.org/10.7717/peerj.5765#supplemental-information.

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
