# Peer review of "Building interpretable models for polypharmacy prediction in older chronic patients based on drug prescription records"

_PeerJ, doi:10.7717/peerj.5765_

## Round 0.1 · original submission · Major Revisions

Dear authors,

Your manuscript needs some modifications before being considered for publication. Please, address the comments indicated by the reviewers to improve your paper.

With respect and warm regards,
Dr Palazón-Bru (academic editor for PeerJ)

·

Basic reporting

Firstly, I congratulate to the authors! I consider that they had done a great work! Thank you for giving me the opportunity to revise this manuscript. I found this paper particularly interesting in terms of not only your focus of utility on an interpretable model of polypharmacy but in relation to any older chronic patients.

The authors present the differing aspects of your research in an appropriate way and this allows the reader to follow the study and understand the approaches, analysis and discussion of findings which are embedded within the aim of to build models for predicting polypharmacy for chronic patients.

I appreciate you will have put in a lot of effort in preparing your paper and there are a number of points which I offer to assist with further developing the manuscript. This points are listed under the subheadings.

Introduction:
1. In line 19 appears the cite number “20” and in the line 40 appears the cite number “33”. I suppose that this is an error. Please, update the cites in order of apparition.
2. Why did you focus on chronic cardiovascular disease and type 2 diabetes mellitus? Are they the most frequent? Are they the more associated to polypharmacy? I suggest that you explain it in the introduction section.

Methodology:
3. In line 50 appears “The first dataset contained drug prescription records collected in Slovenia from 2008 to 2016” Can the authors explain what type of records contained? Did they belong to primary care, hospital or both?

References:
4. Please, check the references! There are some references cited incorrectly.
5. I think that your literature should be the most recent on this topic; there are 12 of 34 references belong to before 2012. Try to include the most up to date international literature!

Experimental design

You are building models for polypharmacy prediction in older chronic patients with a large sample of Slovenia older people, being very interesting for PeerJ readers.

The authors described with sufficient details all methods utilized.

Validity of the findings

I appreciate you will have put in a lot of effort in preparing your paper and there are a number of points which I offer to assist with further developing the manuscript. This points are listed under the subheadings.

Results:
1. It would be interesting the authors offers a supplementary table with the demographic data of the participants.

Discussion:
2. I suggest that you contrast your finding with more international literature.
3. It would be interesting that the authors explain the implications of this study in the Healthcare system.

Conclusion:
4. Could you identify future directions?

Reviewer 2 ·

Basic reporting

no comment

Experimental design

no comment

Validity of the findings

no comment

Additional comments

First of all, I would like to congratulate the authors for the article that I find very interesting and important for the readers of PeerJ and for society in general, because with this prediction model can benefit the health of many people.
I am pleased with the opportunity to review the article and be able to contribute to its improvement in the observation of some data that could have created any doubt after reading it and could clarify me. I will detail them by sections:

Introducction:
It is observed that the citation does not follow a numerical order, since for example on line 40 it appears quote 33, on line 29 it appears quote 20 when it would not be the number that would correspond according to its order of citation

Methods:
- Is there a reason why the date of the data of the discharges included in the article is 2 years before the prescription data?
- What is the criterion that the "unique patients" have to have since from the total sample of 94,475,895 the unique patients are only 755,966.
- If the hospital discharges are a total of 174,061, how is it that 526,087 of unique patients are obtained? Would these patients be included in the 755,966? If they are included, what criteria were carried out to obtain these figures?
- In line 73-74, it shows that the data were different between 2006-2012 and 2013-2016 and used only the 2013-2016 data. The data from 2006 to 2012 are not taken into account? If this is the case, it should be clarified that only the data from 2013-2016 are used, as it would be convenient to clarify the total number that makes up the sample analyzed in the present work.
- In the calculation of the year of birth, it is not clear how the use of the date of the prescription can have a reliable result of the year of birth, do they establish some criteria of minimum or maximum age for the prescription of these medications in order to obtain a date of probable more reliable birth?

- On line 120, could it be incomplete?

Discussión:
Could it be that the differences you find with respect to the literature that increases polypharmacy with age is related to the approximate calculation of the birth date of the participants in your study?

---

## Round 0.2 · accepted · Accept

Dear authors,

I am pleased to inform that your paper has been accepted for publication in its current form in PeerJ.

Congratulations!

With respect and warm regards,
Dr Palazón-Bru (academic editor for PeerJ)

·

Basic reporting

The authors have correctly made the proposed revisions

Experimental design

The authors have correctly made the proposed revisions

Validity of the findings

The authors have correctly made the proposed revisions

Additional comments

I congratulate to the authors! There has been a significant rewrite of this paper. The issues of references have been addressed this has positively impacted on the paper´s quality.

Reviewer 2 ·

Basic reporting

Additional information is suitable.

Experimental design

Additional information is suitable.

Validity of the findings

Additional information is suitable.

Additional comments

I would like to congratulate another one the authors for the interesant article